# Calculation Model of Mechanical and Sealing Properties of NiTi Alloy Corrugated Gaskets under Shape Memory Effect and Hyperelastic Coupling: Two Sealing Properties

**DOI:** 10.3390/ma15134659

**Published:** 2022-07-02

**Authors:** Lingxue Zhu, Yang Liu, Mingxuan Li, Xiaofeng Lu, Xiaolei Zhu

**Affiliations:** 1Department of Mathematics, Jinling Institute of Technology, Nanjing 211169, China; zlx1987@jit.edu.cn; 2School of Mechanical and Power Engineering, Nanjing Tech University, Nanjing 211816, China; liuyang-pro@njtech.edu.cn (Y.L.); lmx9989w@njtech.edu.cn (M.L.); xflu@njtech.edu.cn (X.L.)

**Keywords:** sealing property, NiTi alloy corrugated gasket, leakage rate prediction, surface roughness

## Abstract

Bolted flange connections are commonly used in process industries. Their sealing performance is greatly affected by the gasket. In this paper, a NiTi alloy corrugated gasket was simulated to reveal its sealing performance, considering the gasket surface roughness, shape memory effect and superelastic effect. A fluid–structure coupling analysis method that takes the real surface morphology of the gasket contact zone was proposed, and a leakage rate prediction model was established. The results showed that NiTi shape memory effect could enhance the sealing reliability in service and lower the leakage rate. The leakage rate of the NiTi alloy corrugated gasket is positively correlated with the internal pressure of the medium and the roughness of the sealing surface. The prediction model of the NiTi alloy corrugated gasket leakage rate has good reliability with an average error of about 16.81% compared with the simulation.

## 1. Introduction

A bolted flange connection is the most common connection in the process industry. Leakage failure is the main failure mode of bolted flange joints. It is affected by many factors, such as pre-loading force, temperature, pressure, gasket structure and materials, and bolts and flanges’ structural parameters and materials. Each influencing factor has a strong coupling effect, leading to the complex bolted flange joint leakage failure mechanism. The gasket is the core component of the bolted flange connection. Its performance seriously affects the safety of the equipment. In recent years, process equipment has begun to develop towards high parameterization, multi-function, and large scales [1,2], which puts forward higher requirements on the sealing performance of equipment connections. In other words, the sealing components are required to have good environmental adaptability and good reliability so that the leakage of the joint caused by temperature change or load fluctuation can be prevented. However, a study focused on new sealing gasket material or sealing structure cannot meet all the requirements above.

The corrugated metal gasket has been widely used in chemical industry and nuclear energy fields because of its good compressive resilience and high specific pressure of line seal. Shape memory alloy (SMA) is a sealing material with great potential. Its hyperelasticity and shape memory effect can effectively improve the sealing reliability of bolted flange connections under fluctuating load conditions. However, the high hardness of SMA makes it difficult for the SMA flat gasket to give play fully to its advantages [3]. Therefore, combining the structural advantages of corrugated gasket with the material advantages of shape memory alloy is the key to developing high reliability sealing for high parametric equipment.

In engineering applications, the leakage rate is usually used to characterize the gaskets’ sealing performance. The gasket sealing performance calculation model included the gasket leakage rate calculation model and the prediction model of sealing leakage failure. Deltombe et al. [4] established a theoretical leakage model based on roughness through a multi-dimensional roughness analysis of double-acting pumps processed by traditional machining and ultra-precision machining. The results showed that the dual-action pump processed by ultra-precision machining has better sealing performance. Meanwhile, it also showed that the micro-roughness significantly affects the sealing performance of gaskets [5,6,7]. Muhsin et al. [8,9] studied the sealing performance of the convex surface of the metallic bolted flange joint based on the pressure–penetration criterion (PPNC) in ANSYS. A leakage rate prediction model was established. The benchmark test verified the results, and the prediction model was in good agreement with the test. Compared with other leakage rate models, the prediction model based on PPNC to simulate fluid pressure seepage is more accurate and very efficient. It was found that the leakage initially began at the inside of the gasket. In addition, due to the uneven distribution of bolt load in the circumferential direction, leakage growth at the midpoint between bolts is greater than that at the center of bolts. On this basis, Muhsin et al. [10,11,12,13,14] investigated the leakage propagation between the flange and the gasket of bolted fiber-reinforced polymer flange joints using the PPNC. Yu et al. [15,16] established a calculation model of leakage rate based on seepage mechanics. This model successfully predicted the size and leakage rate of the leakage channel on the surface of the mechanical seal, which played a constructive role in the optimization design and improvement of gasket sealing performance. Based on the fractal theory of porous media, Ni et al. [17] successfully established a theoretical model for calculating fluid leakage rate at the interface of mechanical seal by comprehensively considering the actual sealing contact interface. The theoretical results were verified and have the same order as the leakage rate predicted by the conventional model, which indicates that the model has certain feasibility. The model played a certain role in promoting the leakage rate prediction and the future study of the leakage mechanism.

In order to explore the influencing factors of sealing performance of the new corrugated gaskets, the numerical analysis method and theoretical leakage prediction model of hard sealing based on roughness were studied. Haruyama et al. [18] studied the influence of roughness on sealing performance, and the test results showed that roughness had a significant impact on leakage rate under low pre-load force. Based on the penetration theory, Ma et al. [19] established the theoretical leakage model of a gasketless bolted flange joint with full consideration of the surface roughness. The model uses the Persson contact theory to predict the critical shrinkage height along the penetration path to give the maximum allowable bolt spacing to ensure a low leakage rate. After finite element analysis and demonstration, the analysis results are consistent and the accuracy is high. Ji et al. [20] used the Hertz theory to conduct relevant research on the contact mechanics of a mechanical seal interface and established the sealing surface leakage channel model, which demonstrated the relationship between the leakage rate and the surface topography. The predicted value of the leakage rate theoretical model was in good agreement with the test results. Huang et al. [21] introduced the fractal porous media theory into the study of the leakage mechanism of metal gaskets and established a theoretical prediction model of the leakage rate of metal gaskets. The results showed that the predicted value is consistent with the measured value and the gasket’s roughness significantly influences the leakage rate [22]. Based on the relationship between hard metal’s roughness and leakage, Peng et al. [23] proposed the magnification-based theory and multi-asperity theory. They established theoretical leakage models based on the idea above. It was found that the magnification-based model is more consistent with the experiment, but the multi-asperity model is more suitable for engineering practice.

In conclusion, for metallic gaskets, surface roughness has a significant influence on sealing performance. Scholars have conducted a lot of research on the gasket leakage rate prediction model, gasket surface roughness quantitative characterization method, and the influence of roughness on gasket sealing performance. However, in the research of the gasket sealing performance based on the roughness of gasket, rough surfaces were generated by GUASS statistical characteristic equation, metal bumps were generated by the simplified model, and the coupling between bumps was not considered. All these factors made the calculation results of the prediction model deviate from the experimental results. At the same time, there are few studies on sealing performance of NiTi alloy corrugated gaskets.

In this paper, based on the 3D geometric reconstruction method, a geometric model modeling method of NiTi alloy corrugated gasket with the real surface state was proposed. On this basis, a simulation analysis model for the leakage process of NiTi alloy corrugated gasket with a real surface state was established, combined with the analysis method of fluid–structure interaction. The influence of contact stress and roughness on the leakage characteristics of NiTi alloy corrugated gaskets was explored. According to the simulation analysis results, a leakage rate prediction model of NiTi alloy corrugated gasket was established.

## 2. Materials and Methods

### 2.1. Theoretical Model

Compared with the traditional metal flat gasket, the corrugated gasket is a typical multi-channel seal. As shown in Figure 1, the crest and trough of the corrugated gasket form sealing faces with the upper and lower flanges, respectively. In other words, the corrugated gasket forms three sealing faces with the upper flange and four sealing faces with the lower flange. Each sealing face presents a concentric ring plane. Without considering the influence of the gasket surface roughness on the sealing performance, each sealing face can be simplified into a parallel circular plate, and the following equation can calculate its leakage rate:(1)$$Q_{r}=\frac{\pi h^{3}\left(p_{1}-p_{2}\right)}{6\mu \cdot \ln\left(d_{2}/d_{1}\right)}$$
where *h* (m) is the clearance height of the sealing face; *p*_1_ (Pa) and *p*_2_ (Pa) represent the pressure inside and outside the sealing face, respectively; *d*_1_ (m) and *d*_2_ (m) denote the inner and outer diameter of the sealing face, respectively; and *μ* (Pa·s) is the dynamic viscosity of the internal fluid.

There are various machining marks on the metal surface for the actual service sealing gasket, making the sealing surface appear to have irregular rises and pits. The unsmooth sealing face with bumps and pits comes into contact with the flange, forming connected porous interface areas, as shown in Figure 2. These connected porous areas naturally form the leakage channels of the fluid medium. Therefore, considering the influence of gasket surface roughness on gasket sealing performance, the gasket leakage rate can be calculated by the following equation:(2)$$Q=K\cdot Q_{r}=\frac{K\pi h^{3}\left(p_{1}-p_{2}\right)}{6\mu \cdot \ln\left(d_{2}/d_{1}\right)}$$
where *K* is the leakage factor. As shown in Figure 2, driven by the internal pressure, the fluid leaks out of the gap between the gasket sealing face and the flange. The higher the gap height is, the smaller the fluid flow resistance will be, and the higher the leakage rate will be. For the sealing surface with bumps and pits, the gasket surface can hinder the flow of fluid. Therefore, the leakage factor can be calculated by the following equation:(3)$$K=1-e^{A\cdot {\left(h/\sigma \right)}^{B}}$$
where *σ* is the roughness of gasket sealing face. *A* and *B* are constants.

Under bolt load, compressive deformation occurs on the gasket. The bulge on the gasket surface is compressed so that the clearance between the gasket and the flange decreases. With the increase of bolt load, the stress of the gasket increases gradually, and the height of seal clearance decreases gradually. Therefore, the clearance height of the sealing face is directly related to the stress of the gasket, which the following equation can calculate:(4)$$h=3\sigma \cdot e^{C\cdot S_{{}_{G}}^{D}}$$
where *S_G_* is the gasket stress and *C* and *D* are constants. By substituting Equations (3) and (4) into Equation (2), the leakage rate calculation model considering gasket surface roughness can be obtained, as shown in Equation (5):(5)$$Q=\left(1-e^{A\cdot {\left(3\cdot e^{C\cdot S_{G}^{D}}\right)}^{B}}\right)\cdot \frac{\pi {\left(3\sigma \cdot e^{C\cdot S_{G}^{D}}\right)}^{3}\left(p_{1}-p_{2}\right)}{6\mu \cdot \ln\left(d_{2}/d_{1}\right)}$$

### 2.2. Leakage Rate Simulation Method Based on Fluid–Structure Interaction with Real Gasket Sealing Surface Morphology

According to the deformation mechanism of NiTi alloy corrugated gasket, the second ripple of the NiTi alloy corrugated gasket is the area with the greatest contact stress, and it is also the main area where the gasket plays a sealing role. In this paper, the second ripple of the NiTi alloy corrugated gasket is taken as the research object to study its sealing performance. Figure 3 shows the modeling process of the digital geometric model with real gasket sealing surface topography. In the first step, the sealing face of the NiTi alloy corrugated gasket was scanned by an OLYMPUS DX510 microscope with a 3D ultra-depth of field scanning method. In the second step, the scanning curve was discretized and the spline interpolation method was used to smooth the discrete points to obtain the surface contour. The third step was importing the surface contour into the SOLIDWORKS modeling software to generate the surface. In the last step, the Boolean logic operation was performed to create a solid model. Thus, a digital geometric model with real gasket sealing face morphology was obtained, as shown in Figure 3f.

NiTi alloy corrugated gaskets underwent two stages during service. The first stage was the gasket installation stage, also known as the pre-loading stage. The gasket was placed between the two flanges, which compress the gasket under bolt load. During the installation process, NiTi alloy was in the martensite state. Under the bolt load, martensite detwinning occurred. Multiple martensite variants converted to single martensite. Moreover, the NiTi alloy corrugated gasket buckling occurred in the straight edge segment, the corrugated angle became big, and the overall height of the gasket decreased. The flange compressed the bulge of the sealing face, and the sealing face clearance was reduced. The second stage was the operation stage. The gasket of the bolted flange joint deformed under the medium internal pressure and temperature load. Firstly, the two flange surfaces were separated. That is, the gasket was elastically deformed. Secondly, under medium temperature, flange, bolt, and gasket materials soften and expand. At this stage, the austenitic transformation occurred in NiTi alloy. The bulges on the sealing surface were transformed from martensite to austenite. At the same time, the bulges and pits have complex deformation behavior because of thermal expansion.

Due to the NiTi alloy’s serious phase transformation behavior and the severe localization of sealing face deformation, the sealing face deformation of the NiTi alloy corrugated gasket with real sealing face profile features was complicated. It had an important influence on the sealing performance of the gasket. In order to reveal the deformation mechanism of the sealing face of NiTi alloy corrugated gasket with real sealing face profile features, a simulation analysis method was proposed based on ABAQUS/Standard. The maximum distance between the bulge and the pit on the NiTi alloy corrugated gasket sealing face is about 11 μm, which is 3.7% of the thickness of the plate (0.3 mm) and 0.75% of the gasket’s height. The deformation of bulge was mostly local deformation. Therefore, to simplify the calculation, a geometric model was established, as shown in Figure 4a. In order to explore the deformation characteristics of the sealing face considering the NiTi shape memory effect and superelasticity effect, but also to guarantee the convergence of computing, three analysis steps were set. In the first step, the lower loading head was fixed. A small displacement was applied to the upper loading head to form stable contact pairs between the upper and lower loading heads and the gasket surface. The initial temperature was set as 293 K. In the second step, the pre-loading stage, a bigger displacement was applied to compress the gasket, and the initial sealing conditions formed. The temperature field was maintained constant. In the third step, the heating stage, to simulate the gasket operation condition, keeping the relative position of upper and lower loading head unchanged, the temperature was set as 393 K. The deformation behavior of the sealing surface under shape memory effect could then be explored. The upper and lower loading heads were set as rigid bodies, the lower loading head adopted the fixed boundary condition (UX = UY = UZ = URX = URY = URZ = 0). The upper loading head adopted the displacement constraint (UX = UZ = URX = URY = URZ = 0). That is, the loading head could compress the gasket along the Y direction. The contact surface between the upper and lower loading heads and the sealing face adopted the penalty function contact, and the friction coefficient was 0.15. The mesh of the NiTi alloy corrugated gasket simplified model with real sealing face profile features was C3D8 element. The mesh size was 40 μm. There were at least two layers of element along the thickness direction in some irregular area. Its mesh size was 10 μm. The mesh model is shown in Figure 4b. To calculate the deformation process of the bulges and pits on the surface of NiTi alloy and obtain the fluid medium leakage micro-channel of the NiTi alloy corrugated gasket, the deformation process was simulated and analyzed by ABAQUS. The material parameters of the NiTi alloy are shown in Table 1, and its constitutive model and UMAT subroutine refer to the literature [24].

The NiTi alloy corrugated gasket’s sealing face deformed under the bolt load so that the bulge and pit were locally deformed. The height of the clearance between the sealing surfaces was reduced. So, the flow resistance of internal fluid medium in the clearance was increased. In order to explore the influence of gasket surface roughness on sealing performance in service, a leakage analysis simulation method of NiTi alloy corrugated gasket with real sealing surface profile features was established based on the Euler–Lagrange coupling analysis method. The calculated deformation results of the NiTi alloy corrugated gasket with real sealing face contour features were imported into ABAQUS/Explicit in the form of a mesh model as the geometric model, as shown in Figure 5. Meanwhile, the Euler domain model of the fluid medium was constructed. In the calculation process, the effect of the internal pressure of the medium on the loading head and the gasket was ignored. The upper loading head and the gasket were set as analytic rigid bodies. It was assumed that the fluid in the Euler domain was nitrogen, and its physical parameters are shown in Table 2. The boundary conditions were shown in Figure 6. The upper loading head and gasket were constrained by fixed support (Fix1, Fix2, UX = UY = UZ = URX = URY = URZ = 0). Fluid in the Euler domain used Outflow with pressure inlet boundary conditions. The upper and lower surfaces of the Euler domain adopted Velocity boundary conditions. The Velocity perpendicular to the upper and lower surfaces is set to 0 (V3 = 0). Similarly, Velocity boundary 2 was set on the left and right surfaces of the Euler domain, i.e., V1 = 0. The element type was EC3D8R with a mesh size of 1 μm.

## 3. Results and Discussion

### 3.1. Influence of Gasket Stress on Sealing Performance

When the surface roughness was 11 μm, the stress and contact stress contours under different loading displacements were shown in Figure 7. With the increase of loading displacement, gasket stress and contact stress gradually increased and the contact area gradually increased. The porous area enclosed by the gasket and flange gradually decreased. Meanwhile, the stress distribution of the gasket presented prominent localized characteristics.

When the loading displacement was 6 μm, the internal pressure load was 5 MPa, and the surface roughness was 11 μm, the flow change curve of the sealing face inlet is shown in Figure 8. The distributions of the medium on the sealing surface at different times are shown in Figure 9. As shown in Figure 8, with the time increasing, the flow rate at the gasket inlet showed a trend of first increasing and then gradually decreasing. As demonstrated in Figure 9, the fluid medium flowed into the porous area surrounded by gaskets and flanges. As the porous area had a significant effect on the obstruction of the fluid medium, the inlet flow rate gradually decreased. When the time reached 1380 μs, the fluid medium filled the porous area of the sealing surface so that the inlet and outlet flow reached equilibrium. Here, the flow rate was the leakage rate of the NiTi alloy corrugated gasket, considering the roughness of the gasket surface. According to the flow characteristics, the gasket leakage rate and contact stress under different loading displacements were calculated, as shown in Figure 10.

As shown in Figure 10, the contact stress on the gasket surface presented a nonlinear increasing trend with the loading displacement. This was because the transformation processes, such as martensite detwinning transformation and austenite transformation, and the deformation characteristics, such as martensite elastic deformation, austenite elastic deformation, and local plastic deformation, occurred during the pre-loading and operation of the NiTi alloy gasket. All the factors above made the mechanical property response of the gasket show strong nonlinear characteristics. With the displacement increasing, the NiTi alloy had more phase transformation and the stress increased.

The gasket leakage rate decreased gradually with the loading displacement increasing. With the increase of the displacement, the porous area enclosed by the gasket and flange gradually decreased. The section of the leakage channel gradually decreased so that the gasket leakage rate gradually decreased. When the loading displacement was less than 8 μm, the leakage rate decreases rapidly. When the loading displacement was larger than 8 μm, the decay rate of leakage rate slowed down. Figure 7 showed that when the loading displacement reached 8 μm, the outlet of the leakage channel in the sealing area was blocked because of the compression of the flange on the gasket surface and formed an effective seal.

### 3.2. Influence of Medium Internal Pressure on Sealing Performance

When the loading displacement was 8 μm and the surface roughness was 11 μm, the gasket leakage rate changed under different internal pressures, as shown in Figure 11. As shown in Figure 11, the relationship between medium internal pressure and gasket leakage rate presented an approximately linear monotonically increasing curve. The fluid medium leakage channel was the same. With the increase of medium pressure, the difference between the pressure inside and outside the gasket sealing surface increased. The leakage driving force inside the sealing surface increased, resulting in more fluid flowing through the sealing surface and increasing the gasket leakage rate.

### 3.3. Influence of Roughness on Sealing Performance

When the flange loading displacement was 8 mm and the medium internal pressure was 5 MPa, the rules of contact stress and leakage rate with different sealing surface roughness changed, as shown in Figure 12. Under the same loading displacement condition, with the increase of surface roughness, the contact stress of the gasket sealing surface gradually decreased and the gasket leakage rate gradually increased. However, when the roughness was 11.58 μm and 15.49 μm, the contact stress of sealing surface presented an opposite law. This is because the flatness of the sealing surface of Model 1 (roughness 15.49 μm) is poor, and the contact surface presented high at one end and low at the other. During the loading process, the high end contacted first, and the contact stress increased with the loading, as shown in Figure 13. Model 2 (roughness of 11.58 μm) has a higher flatness of the sealing surface, with only a few rough peaks higher than the average datum surface. During the flange loading process, the rough peaks come into contact with the flange, while most of the datum surface did not reach the flange, resulting in the small contact stress of the sealing surface. By comparing the relationship between the contact stress and leakage rate of Model 1 and Model 2, it was found that the contact stress of Model 1 was higher than that of Model 2. The leakage rate of Model 1 is lower than that of Model 2, which was inconsistent with the rule mentioned above. This was because there is a great difference in the flatness of the gaskets in Model 1 and Model 2. The leakage channel formed by the contact surface and flange surface of Model 1 was relatively regular and the resistance loss of medium flow in the leakage channel was small. However, the contact surface of Model 2 and the flange formed a complex porous medium area. There were multiple rough peaks in the leakage channel that hindered the flow of the medium, resulting in large resistance of the medium during the flow of the leakage channel. This made the contact stress of Model 2 lower than that of Model 1. In contrast, the leakage rate in Model 2 was lower than that of Model 1.

### 3.4. Prediction Model for Leakage Rate of NiTi Alloy Corrugated Gaskets

The simulation method based on fluid–structure coupling with the real gasket sealing surface topography was adopted to calculate the leakage rate of NiTi alloy corrugated gasket with different loading displacements, medium pressures, and surface roughnesses. The results are shown in Table 3. According to the data in Table 3, Equation (5) was fitted and the prediction model of the leakage rate of NiTi alloy corrugated gasket, considering the shape memory and super-elasticity effects, is shown in Equation (6). The comparison between finite element results and prediction results is shown in Figure 14. As shown in Figure 14, the errors of Model 1, Model 2, and Model 8 were large, with the maximum error reaching 39%. The error of other models was small, and the minimum error was 0.002283%. There may be two reasons for this. First, the error generated by the finite element simulation results and the fitting error of the prediction model was superimposed under the conditions of low pre-loading stress. Second, the prediction model lacked the parameters of the flow resistance characteristics in the contact area of the sealing surface, resulting in a large difference between the calculated leakage rate and the finite element simulation results under low pre-load. The average error between the prediction model and the finite element simulation results was 16.81%, indicating that the prediction model had a certain reliability and met engineering applications’ needs.
(6)$$Q=\left(1-e^{-0.003029\cdot {\left(3\cdot e^{-1.017\cdot S_{G}^{0.1385}}\right)}^{-2.64}}\right)\cdot \frac{\pi {\left(3\sigma \cdot e^{-1.017\cdot S_{G}^{0.1385}}\right)}^{3}\left(p_{1}-p_{2}\right)}{6\mu \cdot \ln\left(d_{2}/d_{1}\right)}$$

## 4. Conclusions

In this paper, the prediction model of leakage rate of the NiTi alloy corrugated gasket with narrow sealing surface was studied. The simulation method of leakage rate with real gasket sealing surface morphology was established based on the fluid–structure coupling method. The influences of the NiTi alloy shape memory effect, superelasticity, medium internal pressure, and gasket surface roughness on the sealing performance were discussed. The prediction model of the leakage rate of the NiTi alloy corrugated gasket was established. The conclusions are as follows:(1)With the increase of loading displacement, NiTi shape memory effect is fully exerted. This causes the contact stress of the sealing surface to show a nonlinear increase. The gasket leakage rate gradually decreases. When the loading displacement is 8 μm, the NiTi alloy corrugated gasket has good sealing performance.(2)The leakage rate of the NiTi alloy corrugated gasket is positively correlated with the internal pressure of the medium and the roughness of the sealing surface. The greater the internal pressure of the medium and the greater the surface roughness, the greater the leakage rate.(3)The prediction model of the NiTi alloy corrugated gasket leakage rate was established. The average error was 16.81% compared with the finite element simulation results.

## Figures and Tables

**Figure 1 materials-15-04659-f001:**
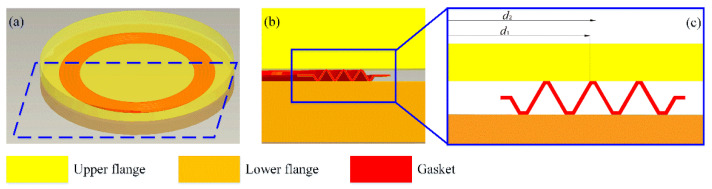
Schematic diagram of corrugated gasket seal: (**a**) corrugated gasket and flangs, (**b**) section of corrugated gasket, (**c**) partial enlarged detail.

**Figure 2 materials-15-04659-f002:**
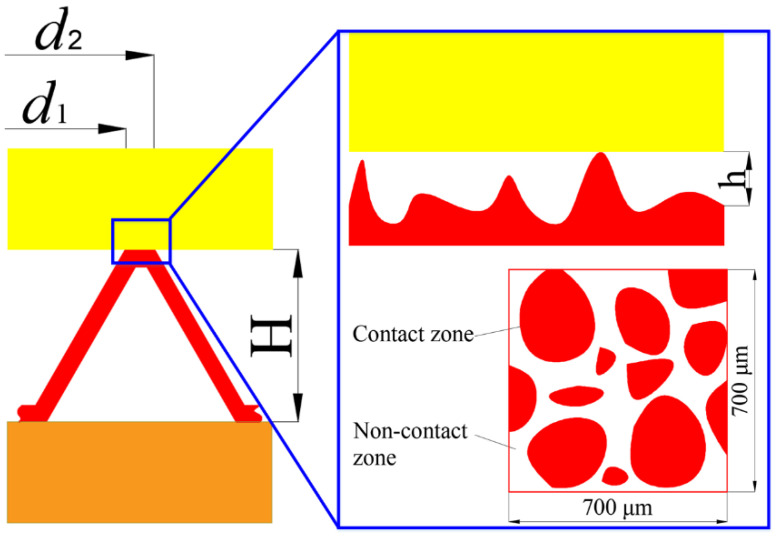
Schematic diagram of the porous interface on real sealing face.

**Figure 3 materials-15-04659-f003:**
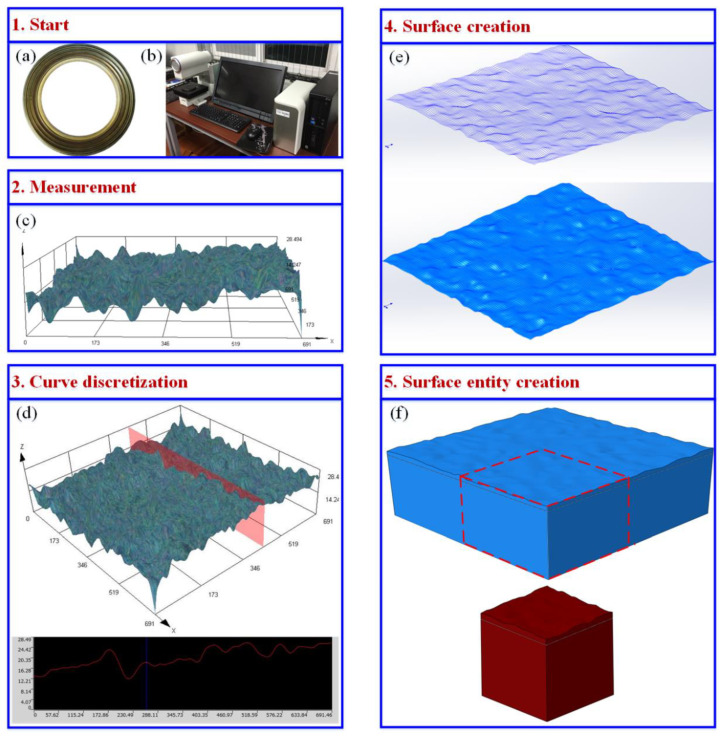
Digital geometric model modeling process with real sealing face topography: (**a**) corrugated gasket, (**b**) OLYMPUS DX510 microscope, (**c**) measurement of the gasket surface, (**d**) curve discretization, (**e**) surface creation, (**f**) surface entity creation.

**Figure 4 materials-15-04659-f004:**
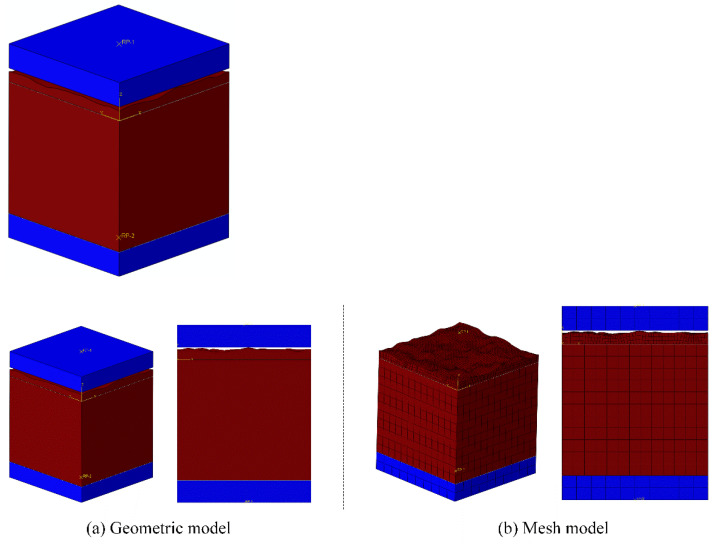
Finite element model.

**Figure 5 materials-15-04659-f005:**
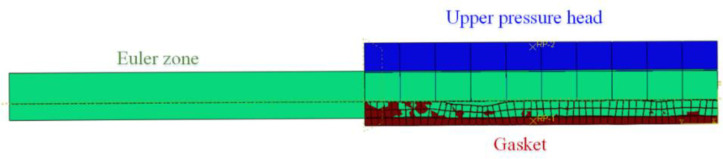
Geometric model of Euler-Lagrange coupling analysis method.

**Figure 6 materials-15-04659-f006:**
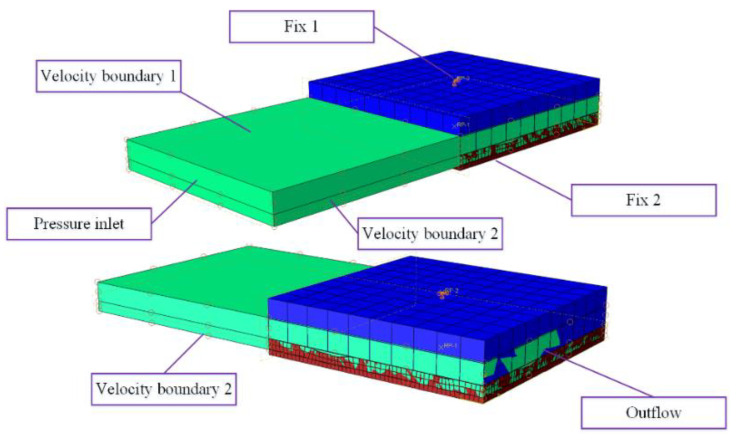
Schematic diagram of boundary conditions.

**Figure 7 materials-15-04659-f007:**
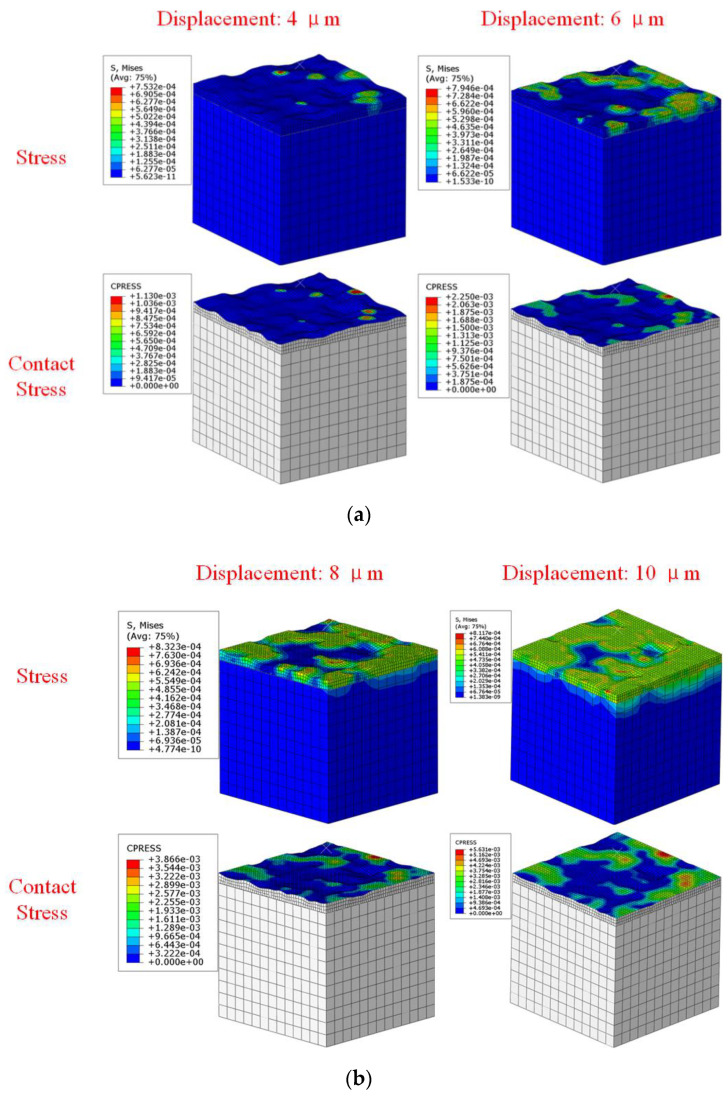
Stress and contact stress contour of gasket contact area under different loading displacement conditions (unit of stress and contact stress is N/μm^2^): (**a**) displacement 4 μm and 6 μm, (**b**) displacement 8 μm and 10 μm.

**Figure 8 materials-15-04659-f008:**
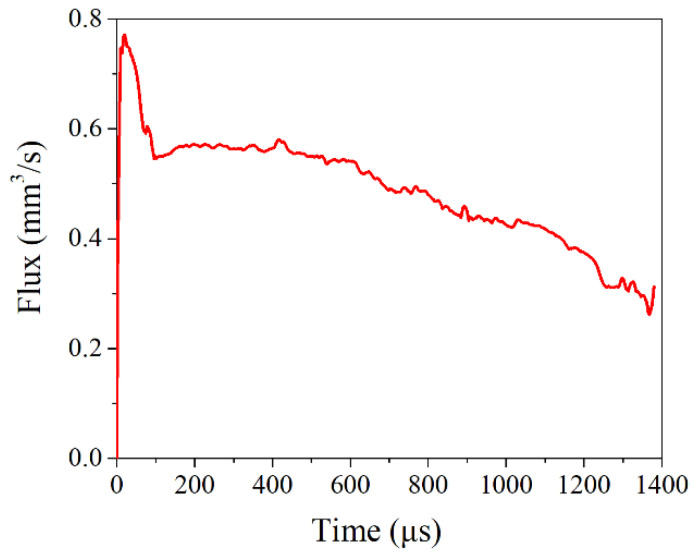
Inlet flow variation curve of sealing surface.

**Figure 9 materials-15-04659-f009:**
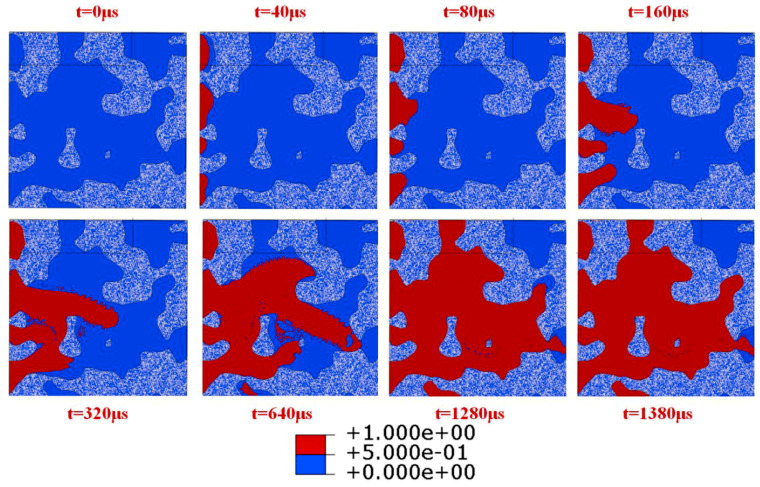
Distribution characteristics of medium on sealing surface.

**Figure 10 materials-15-04659-f010:**
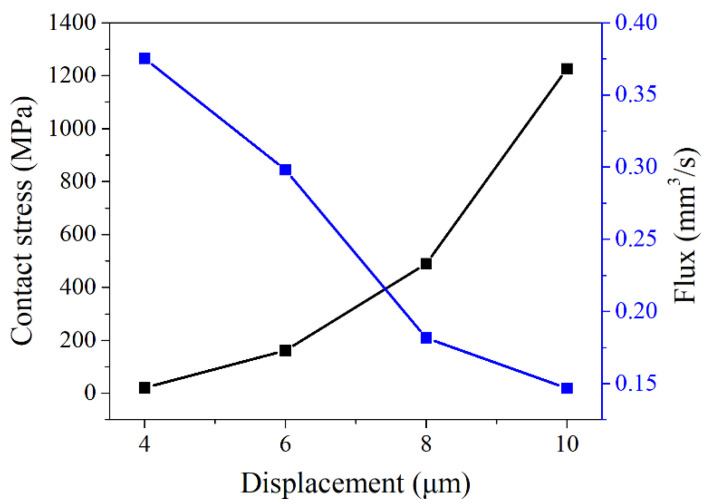
The relationship between the loading displacement and the contact stress and leakage rate of gasket.

**Figure 11 materials-15-04659-f011:**
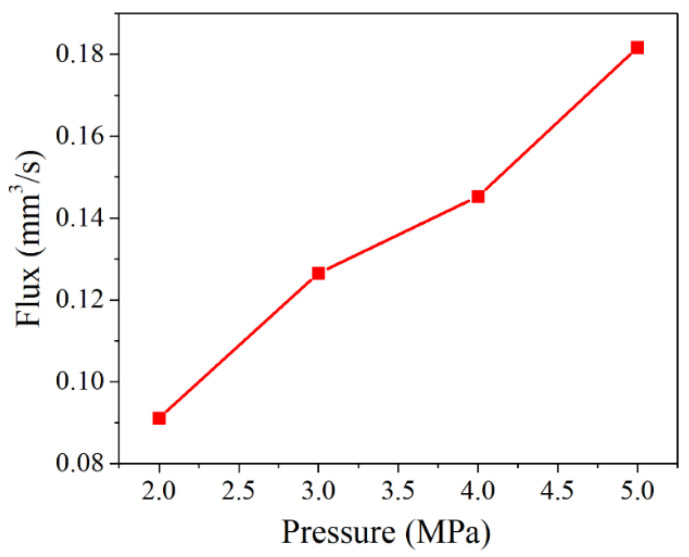
The relationship between the internal pressure of medium and leakage rate.

**Figure 12 materials-15-04659-f012:**
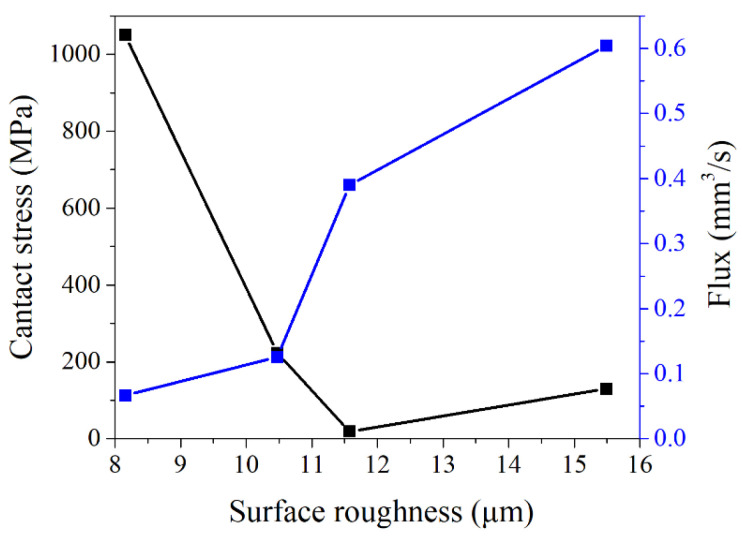
The relationship between surface roughness and contact stress and leakage rate.

**Figure 13 materials-15-04659-f013:**
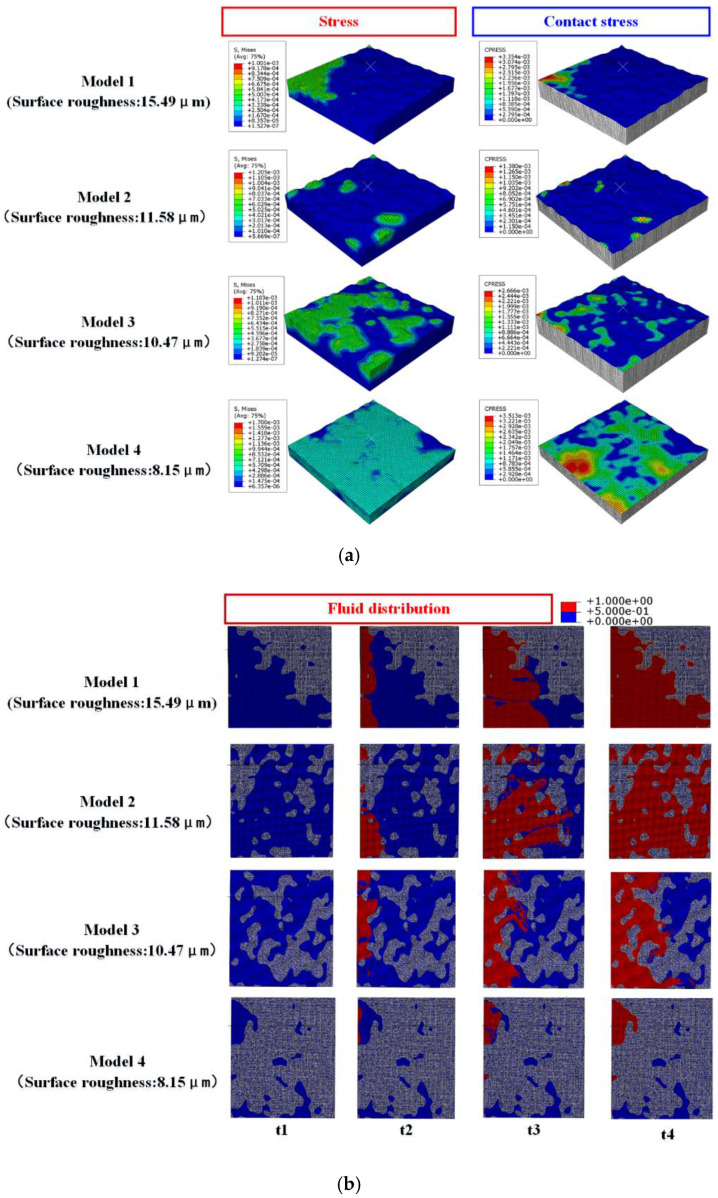
The gasket stress, contact stress, and medium flow characteristics of different surface roughness models: (**a**) gasket stress, contact stress distribution, (**b**) medium flow characteristics.

**Figure 14 materials-15-04659-f014:**
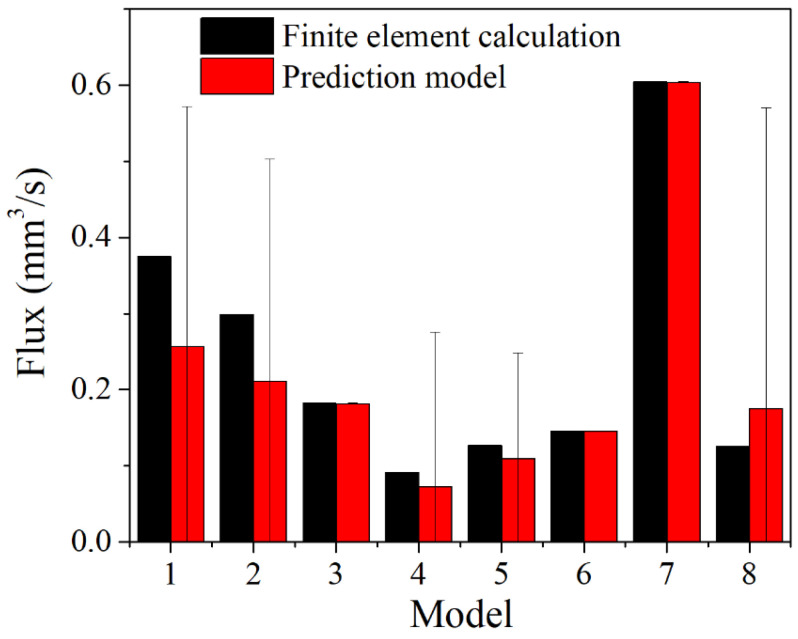
The comparison between calculation results and model prediction results.

**Table 1 materials-15-04659-t001:** Material parameters of the NiTi alloy.

Physical Quantities	Parameter Selection
Density/(g/cm^3^)	6.45
Young’s modulus of pure martensite/GPa	45
Young’s modulus of pure austenite/GPa	61
Poisson’s ratio	0.33
Martensite influence coefficient (MPa/K)	15.8
Austenite influence coefficient (MPa/K)	15.8
Maximum residual strain	0.023
Plastic limit/MPa (120 °C)	521
Plastic limit/MPa (20 °C)	618

**Table 2 materials-15-04659-t002:** Physical parameters of nitrogen.

Physical Quantities	Parameter Selection
Density (kg/μm^3^)	1.138 × 10^−18^
Critical speed (μm/s)	4.2081 × 10^8^
Kinematic viscosity (N·s/μm^2^)	1.1663 × 10^−11^

**Table 3 materials-15-04659-t003:** Calculation results of the leakage rate of the NiTi alloy corrugated gasket under different parameters.

Number	Medium Internal Pressure/MPa	Inner Diameter of Sealing Surface/mm	Outer Diameter of Sealing Surface/mm	Roughness/μm	Contact Stress/MPa	Leakage Rate/mm^3^/s
1	5	100.69	100.99	11	21.0352	0.3752
2	5	100.69	100.99	11	161.682	0.2981
3	5	100.69	100.99	11	489.48	0.1817
4	2	100.69	100.99	11	489.48	0.09111
5	3	100.69	100.99	11	489.48	0.12653
6	4	100.69	100.99	11	489.48	0.14527
7	5	100.69	100.99	15.48716	130.14	0.60428
8	5	100.69	100.99	10.47273	223.372	0.12555

## Data Availability

Followed according to MDPI Research Data Policies.

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
