# Peer review of "Calculation Model of Mechanical and Sealing Properties of NiTi Alloy Corrugated Gaskets under Shape Memory Effect and Hyperelastic Coupling: Two Sealing Properties"

_materials, 2022, doi:10.3390/ma15134659_

Round 1

Reviewer 1 Report

·           The link between Section 2 and 2.1 is missing.

·           Sources of Equations needed, add ref.

·           Add all research objectives after the Introduction as appearing in 157-158.

·           Table 1 adds a reason for the selection of parameters.

·           The figures are well drawn. However, the quality of Figures 8 and 10 can be enhanced.

·           What is the significance of error analysis? First, it can be added or add error bars in Figures 9, 10, 11, 13, etc., as Added in Fig. 15 looks fine.

·           An explanation of Table 3 is missing in the manuscript. Therefore, add a contribution to Table 3.

·           The significance of Fluid-Structure coupling methods is missing support from Literature.

·           The prediction model of NiTi alloy corrugated gasket leakage rate was established. And the average error was 16.81% compared with the finite element simulation results. Comparison with the Literature is missing; justify it (Model).

·           The model has good Reliability—conclusion 3 (What is the source of the paper results regarding Reliability).

·           What type of Reliability was checked for the model? Unfortunately, I don’t find any information that appears in the paper.

§   Concluding remarks can be improved. Write down managerial implications. Conclusions need revision.

·           Abstract also need improvement. Authors are requested to spend time. Future scope addition can enhance quality.

Author Response

  1. The link between Section 2 and 2.1 is missing.

The authors' Answer:

We sincerely appreciate the valuable comments. In the second chapter, we put together the leakage rate calculation theoretical model of corrugated gaskets based on parallel circular plates and the simulation analysis model for the leakage process of NiTi alloy corrugated gasket based on fluid-structure interaction. It may make it difficult for readers to understand the logic of this chapter. Therefore, we separate the two parts. The second chapter is the leakage rate calculation theoretical model of NiTi alloy corrugated gasket. The third chapter is the simulation analysis method for the leakage process of NiTi alloy corrugated gasket based on fluid-structure interaction.

  1. Sources of Equations needed, add ref.

The authors' Answer:

We sincerely appreciate the valuable comments. The leakage rate calculation model of the NiTi alloy corrugated gasket proposed in this paper is based on a parallel circular flat plate. The model considers the influence of the NiTi alloy corrugated gasket surface morphology and the leakage microchannel formed by the flange surface on the gasket sealing performance. The leakage rate calculation method proposed in this paper is a simplified method suitable for engineering. The Formula (1) is the most common Formula for calculating the leakage rate of metal gaskets. It has been written into textbooks, so no references are written. The Formula (2)-(5) are all amendments to Formula (1) proposed by the author.

  1. Add all research objectives after the Introduction as appearing in 157-158.

The authors' Answer:

We sincerely appreciate the valuable comments. The original text is modified as follows:

"In this paper, based on the 3D geometric reconstruction method, a geometric model modeling method of NiTi alloy corrugated gasket with the real surface state was proposed. On this basis, a simulation analysis model for the leakage process of NiTi alloy corrugated gasket with a real surface state was established, combined with the analysis method of fluid-structure interaction. The influence of contact stress and roughness on the leakage characteristics of NiTi alloy corrugated gaskets was explored. According to the simulation analysis results, a leakage rate prediction model of NiTi alloy corrugated gasket was established."

  1. Table 1 adds a reason for the selection of parameters.

The authors' Answer:

We sincerely appreciate the valuable comments. The NiTi alloy corrugated gaskets have bulges and pits of varying height and depth on their surfaces. The bulges will be compressed under the preload of the bolt. And as the preload increases, the degree to which the bulges are compressed increases. Therefore, the contact stress of the gasket increases with the increase of preload, and the gasket leakage rate decreases. However, although the bulges on the surface of the NiTi alloy corrugated gasket are compressed, and the bulges still have a certain height, which constitutes the leaking micro-channels. The research on the flow characteristics of the medium in the microchannel helps establish the leakage rate calculation model of the NiTi alloy corrugated gasket. Therefore, this paper studies the leakage process of NiTi alloy corrugated gasket using the fluid-structure coupling method. In the solid domain computing module, the material parameters of NiTi alloy (the parameters in Table 1) are required to explore the formation process of micro-channels in the real surface state. The material parameters of the fluid medium (Table 2) are required in the fluid calculation section. Therefore, this article is modified as follows:

"To calculate the deformation process of the bulges and pits on the surface of NiTi alloy and obtain the fluid medium leakage micro-channel of NiTi alloy corrugated gasket, the deformation process was simulated and analyzed by ABAQUS. The material parameters of NiTi alloy are shown in Table 1, and its constitutive model and UMAT subroutine refer to the literature [23]."

  1. The figures are well drawn. However, the quality of Figures 8 and 10 can be enhanced.

The authors' Answer:

We sincerely appreciate the valuable comments. We have changed the save format of the figures and enhanced the quality.

  1. What is the significance of error analysis? First, it can be added or add error bars in Figures 9, 10, 11, 13, etc., as Added in Fig. 15 looks fine.

The authors' Answer:

We sincerely appreciate the valuable comments. Figure 9 shows the relationship between the flow rate and time of the fluid medium at the inlet end of the leaking microchannel formed by the NiTi alloy corrugated gasket and the flange, which is used to illustrate the hindering effect of the microchannel structure on the fluid flow. Figure 10 shows the fluid flow characteristics in the microchannel at different times. The two figures are calculated by the simulation analysis model for the leakage process of the NiTi alloy corrugated gasket with the real surface state to illustrate the flow characteristics in the microchannel, and there is no error comparison. Figure 11, Figure 12, and Figure 13 express the influence of various factors that affect the sealing performance of the gasket on the leakage rate. These results are obtained from simulation analysis and are not compared with experiments or other theories, so there is no error analysis. Figure 15 compares the leakage rate obtained by the theoretical calculation model and the calculation result of the finite element simulation analysis. Therefore, the figure is subjected to error analysis.

  1. An explanation of Table 3 is missing in the manuscript. Therefore, add a contribution to Table 3.

The authors' Answer:

We sincerely appreciate the valuable comments. This is the author's writing error. Table 2 on line 339 of the original text should be Table 3. Modifications have been made in the original text.

  1. The significance of Fluid-Structure coupling methods is missing support from literature.

The authors' Answer:

We sincerely appreciate the valuable comments. The fluid-solid coupling analysis method mainly simulates the interaction between fluid and solid. This method is widely used in engineering practice, such as the influence of liquid sloshing on solid deformation. This paper uses this method to calculate the leakage characteristics of NiTi alloy corrugated gasket, a more advanced common method to calculate the leakage rate of gaskets. Based on this method, we use the 3D geometric reconstruction method to obtain the geometric model of the real topographic features, which belongs to the supplement and improvement of this method. Therefore, we believe that references are not required to be cited here.

  1. The prediction model of NiTi alloy corrugated gasket leakage rate was established. And the average error was 16.81% compared with the finite element simulation results. Comparison with the Literature is missing; justify it (Model).

The authors' Answer:

We sincerely appreciate the valuable comments. Usually, the error of simulation analysis mainly comes from the accuracy of the constitutive model and the error caused by the numerical solution method. When the calculation method is compared with the experiment, if the error is less than 20%, it can be considered that the simulation analysis results can replace the experimental results and provide guidance for practical engineering problems. In addition, the research on the test experiment of NiTi alloy corrugated gasket sealing performance has not been reported in the literature. Our team is also working hard to build an experimental device for this work, so there is no comparison with the literature and experiments.

  1. The model has good Reliability—conclusion 3 (What is the source of the paper results regarding Reliability).

The authors' Answer:

We sincerely appreciate the valuable comments. Usually, the error of simulation analysis mainly comes from the accuracy of the constitutive model and the error caused by the numerical solution method. When the calculation method is compared with the experiment, if the error is less than 20%, it can be considered that the simulation analysis results can replace the experimental results and provide guidance for practical engineering problems. For this paper, the constitutive model used has been verified by other students in the research group, and the maximum error is 11.2%, which can guarantee the correctness of the calculation results. The error calculated by applying this model in this paper is 16.1%, which does not exceed 20%, so the author of this paper considers it credible. In order not to make readers doubt this place, we have revised the conclusion:

"(3) The prediction model of NiTi alloy corrugated gasket leakage rate was established. And the average error was 16.81% compared with the finite element simulation results."

  1. What type of Reliability was checked for the model? Unfortunately, I don't find any information that appears in the paper.

The authors' Answer:

We sincerely appreciate the valuable comments. Regarding the correctness of model calculation, the first is the correctness of simulation analysis. For this paper, the solid deformation and fluid flow characteristics of NiTi alloy corrugated gaskets were calculated using the fluid-structure interaction method. For solid analysis, the factors affecting its reliability are the correctness of the constitutive model and the accuracy of the material parameters. We compared the experimental results in another article " Calculation model of mechanical and sealing properties of NiTi alloy corrugated gaskets under shape memory effect and hy-perelastic coupling: â…  Mechanical properties " and in the literature [23], proving the reliability of the calculation. For the calculation of flow characteristics, due to the fluid flow in the microchannel, the flow pattern is mainly laminar flow. In the existing commercial finite element analysis software (such as ABAQUS, and ANSYS), the accuracy of the calculation is very high. In summary, the correctness of the simulation analysis in this paper can be guaranteed. The second is the reliability of the NiTi alloy corrugated gasket's leakage rate calculation model. This paper compares the results of the leakage rate calculation model with the results of the finite element simulation analysis. The reasons for the error are analyzed, and the average error is 16.1%. Although the calculation results cannot be compared with the experiments, the error does not exceed 20%, and the calculation results can also be used to guide practical engineering problems. To sum up, the finite element simulation analysis model and calculation model proposed in this paper have a certain credibility. However, thanks to the reviewer for pointing out the loopholes in this paper, our experimental work is in progress. Due to the high test accuracy requirement of the sensor, the sensor is still under processing. In the follow-up work, we will further confirm the model's credibility through experiments.

  1. Concluding remarks can be improved. Write down managerial implications. Conclusions need revision.

The authors' Answer:

We sincerely appreciate the valuable comments. The conclusions have been revised, and the revised conclusions are detailed in the original text.

  1. Abstract also need improvement. Authors are requested to spend time. Future scope addition can enhance quality.

The authors' Answer:

We sincerely appreciate the valuable comments. We have carefully checked and revised the full text.

Reviewer 2 Report

The paper needs the following revisions for further improvement:

Paragraph-2 of the introduction needs citation of references. Please, also check other sentences which require citation of references. Some references, like [7-13] should be expanded and discussed separately.

Most importantly, the discussion should be expanded with justifications for results.

Author Response

  1. Paragraph-2 of the Introduction needs citation of references. Please, also check other sentences which require citation of references. Some references, like [7-13] should be expanded and discussed separately.

The authors' Answer:

We sincerely appreciate the valuable comments. The Introduction has been modified. The modified part is shown as:

“However, the high hardness of SMA makes it difficult for SMA flat gasket to give full play to its advantages [3].

3 Lu X., Wang C., Li G., Liu Y., Zhu X., Tu S. The Mechanical Behavior and Martensitic Transformation of Porous NiTi Alloys Based on Geometrical Reconstruction. Int J Appl Mech, 2017, 09(03):1750038.”

“Muhsin et al. [8, 9] studied the sealing performance of the convex surface of the metallic bolted flange joint based on the pressure-penetration criterion (PPNC) in ANSYS. And a leakage rate prediction model was established. The benchmark test verified the results, and the prediction model was in good agreement with the test. Compared with other leakage rate models, the prediction model based on PPNC to simulate fluid pressure seepage is more accurate and very efficient. It was found that the leakage began at the inside of the gasket at the earliest. In addition, due to the uneven distribution of bolt load in the circumferential direction, leakage growth at the midpoint between bolts is greater than that at the center of bolts. On this basis, Muhsin et al. [10-14] investigated the leakage propagation between the flange and the gasket of bolted fibre-reinforced polymer flange joints using the PPNC.”

  1. Most importantly, the discussion should be expanded with justifications for results.

The authors' Answer:

We sincerely appreciate the valuable comments. We have carefully revised the full text.

Reviewer 3 Report

Calculation model of mechanical and sealing properties of NiTi 2 alloy corrugated gaskets under shape memory effect and hyper-3 elastic coupling: II sealing properties. 

Lingxue Zhu et al.

In the present manuscript, the authors present a model of mechanical and sealing properties of gaskets prepared from NiTi shape memory alloys. This is an interesting and important application of such shape-memory alloys in industrial applications, thus, the topic is well suited for Materials.

The analysis of the material, i.e., the microstructure, does not play a large role here, which is a pity. However, it might be that as indicated in the title, there is a first part of the analysis/modelling, which may contain such information.

This manuscript comprises 15 figures, 3 tables and 23 references are included.

The manuscript is well prepared and well arranged. The figures, however, have often problems, and especially, too small lettering, which must be changed before publication. Additionally, the figure captions should properly describe what is is shown in the figure. This will help the reader a lot. Furthermore, the authors should consider the possibility to place extra figures into a Supplementary Material.

There are several technical points which require to be addressed prior to publication:

# Fig.1: Resolution of the text is insufficient.

# Fig. 3(d), bottom: The graph looks oly black.

# Figs. 4 and 5 can easily be combined into one.

# Figs. 10 and 14 need to be reworked as the lettering is unreadable.

To summarize, the present manuscript contains interesting material, but it would be useful to include also part I in the documents for refereeing. Then, it would be easier to judge the entire message. In the present form, the manuscript requires a major revision.

Author Response

  1. Fig.1: Resolution of the text is insufficient.

The authors' Answer:

We sincerely appreciate the valuable comments. The figure has been modified.

  1. Fig. 3(d), bottom: The graph looks oly black.

The authors' Answer:

We sincerely appreciate the valuable comments. The figure has been modified.

  1. Figs. 4 and 5 can easily be combined into one.

The authors' Answer:

We sincerely appreciate the valuable comments. The figures have been modified.

  1. Figs. 10 and 14 need to be reworked as the lettering is unreadable.

The authors' Answer:

We sincerely appreciate the valuable comments. The figures have been modified.

Reviewer 4 Report

The manuscript reports on the sealing properties of NiTi alloy corrugated gasket with shape memory effect and hyper elastic coupling via calculation model. The study designed systematically via various theoretical models simulation. The results were coherent and well explained for the sealing performance of NiTi alloy gasket. Therefore, this manuscript is recommended for publication on Materials.

Author Response

We sincerely appreciate the valuable comments.

Round 2

Reviewer 1 Report

Authors revised the paper as per comments, now acceptable.

Reviewer 3 Report

Well performed revision. All points raised by the referees were properly addressed, and suitable changes were made to the manuscript. Thus, the manuscript is now suitable for publication.